# Clonal somatic copy number altered driver events inform drug sensitivity in high-grade serous ovarian cancer

Chromosomal instability is a major challenge to patient stratification and targeted drug development for high-grade serous ovarian carcinoma (HGSOC). Here we show that somatic copy number alterations (SCNAs) in frequently amplified HGSOC cancer genes significantly correlate with gene expression and methylation status. We identify five prevalent clonal driver SCNAs (chromosomal amplifications encompassing *MYC, PIK3CA, CCNE1, KRAS* and *TERT*) from multi-regional HGSOC data and reason that their strong selection should prioritise them as key biomarkers for targeted therapies. We use primary HGSOC spheroid models to test interactions between in vitro targeted therapy and SCNAs. *MYC* chromosomal copy number is associated with in-vitro and clinical response to paclitaxel and in-vitro response to mTORC1/2 inhibition. Activation of the mTOR survival pathway in the context of *MYC*-amplified HGSOC is statistically associated with increased prevalence of SCNAs in genes from the PI3K pathway. Co-occurrence of amplifications in *MYC* and genes from the PI3K pathway is independently observed in squamous lung cancer and triple negative breast cancer. In this work, we show that identifying co-occurrence of clonal driver SCNA genes could be used to tailor therapeutics for precision medicine.

Progressing precision medicine for high-grade serous carcinoma (HGSOC) has been significantly impeded by its significant heterogeneity, driven by chromosomal instability (CIN) resulting in divergent evolution[1,2]. Homologous recombination deficiency (HRD) is the commonest actionable mutational process, notably from germline or somatic deleterious *BRCA1* and *BRCA2* mutations, and routine use of PARP maintenance therapy has significantly extended progression-free survival[3,4]. PARP therapy targets a loss-of-function phenotype by exploiting synthetic lethality effects, but biomarker-driven approaches to identify and target gain-of-function drivers have not been clinically developed.

The frequency of nucleotide substitutions that are "actionable" in HGSOC is ~1% and the vast majority of genomic changes are structural variants, with a high frequency of somatic chromosomal number alterations (SCNAs). Recurrent patterns of SCNAs can be identified across multiple tumours[5,6] as a result of distinct ordering throughout tumour evolution or parallel selection[6–8]. Multiregional analysis across 22 tumour types revealed frequent subclonal focal amplifications in chromosomes 1q (encompassing *BCL9* and *MCL1*), 5p (*TERT*), 11q (*CCND1*), 19q (*CCNE1*) and 8q (*MYC*). *MYC, CCNE1, TERT, KRAS* and genes from the PI3K/AKT/mTOR pathway (e.g. *PIK3CA*) are amongst the commonest amplified cancer genes in HGSOC[2,9–13].

SCNAs frequently drive aberrant gene transcription and protein expression[14,15]. However, SCNA assays are infrequently used in solid tumours. One notable exception is detection of *ERBB2* amplification in breast carcinoma as the critical biomarker for trastuzumab therapy[16,17]. Bulk gene expression profiling in HGSOC has not provided strong evidence of driver gene expression owing to confounding effects from other cells in the tumour microenvironment and genomic heterogeneity[18–20]. By contrast, use of bulk sequencing to detect SCNA, even in low to moderate cellularity cancer specimens, retains high

✉e-mail: f.correiamartins@nhs.net; Charles.swanton@crick.ac.uk; james.brenton@cruk.cam.ac.uk

specificity. In HGSOC, SCNA can be efficiently detected using shallow whole genome sequencing from clinical biopsies[21–24].

Recent clinical trials for HGSOC or other CIN-driven tumours did not show associations between specific mutations and response to therapies[25–27]. We hypothesised that characterization of clonal SCNA in HGSOC might inform drug response and future precision medicine clinical trials to test specific and clinically relevant biomarkers for molecularly targeted therapy. Recent studies using HGSOC organoids and spheroids (cultured and uncultured clusters of primary ovarian cancer cells from HGSOC patient ascites, respectively) suggested that they could be used to predict platinum resistance[28,29]. We here show that spheroids represent the genomic diversity of HGSOC and use them to assess how clonal SCNAs affecting putative HGSOC driver genes correlate with in-vitro drug response.

In this work, we identify prevalent clonal driver SCNAs and demonstrate that they can inform therapeutic response. We show co-occurrence of amplifications in *MYC* and genes from the PI3K pathway in HGSOC and other CIN-driver tumours and that inhibition of the activated mTOR survival pathway in the context of *MYC*-amplified tumours should be considered in future clinical trials (Fig. 1).

## Results

### HGSOC multi-regional analysis shows clonal SCNA driver genes

The GISTIC ("Genomic Identification of Significant Targets in Cancer") algorithm previously defined recurrent somatic copy number alterations (SCNAs) across cancers[30,31]. Although strong associations have been shown previously between chromosomal copy number and gene expression[14,15], it is unclear if these associations are stronger in genes defined as cancer drivers. We hypothesised that SCNAs in

chromosomal segments that include relevant cancer drivers would provide a cellular survival advantage if they were associated with gene expression changes in those relevant drivers and this was not affected by epigenetic silencing. As a consequence, the correlation between SCNA and gene expression would be stronger in cancer drivers than in non-driver genes and SCNAs would not always result in the same gene expression changes due to selection based on methylation status. We first examined the genomic data from the TCGA HGSOC cohort (Fig. 2a; grey shadowing) and found 2415 genes that were affected by amplifications or homozygous deletions in more than 5% of the cases. 156/2415 (6.5%) were putative driver genes based on external curation in the OncoKB Cancer Gene List[13]. These putative SCNA driver genes showed the highest correlation between chromosomal copy number and gene expression, when compared to non-cancer genes and cancer genes from OncoKB with SCNAs in <5% of the cases (Fig. 2b). The same applies when we use different frequency thresholds (Supplementary Figure S1a). They were also more frequently hypomethylated when compared with the remaining genes in TCGA HGSOC cohort (Fig. 2c) and methylation levels were associated with the correlation between chromosomal copy number and gene expression (Fig. 2d).

We further reasoned that frequent SCNA driver events that appear early in tumour progression, here defined as clonal SCNA drivers, should be prioritised as predictive biomarkers for therapy. Therefore, we analysed 127 anatomically distinct HGSOC samples from a cohort of 30 cases with multi-regional sampling in order to time frequent SCNAs in HGSOC evolution[32]. We characterized SCNA as clonal if they were present in all regions from the same individual (median of 4 samples), and subclonal if present in at least one but not all regions. Subclonal SCNAs were very common and distributed across the genome, tending

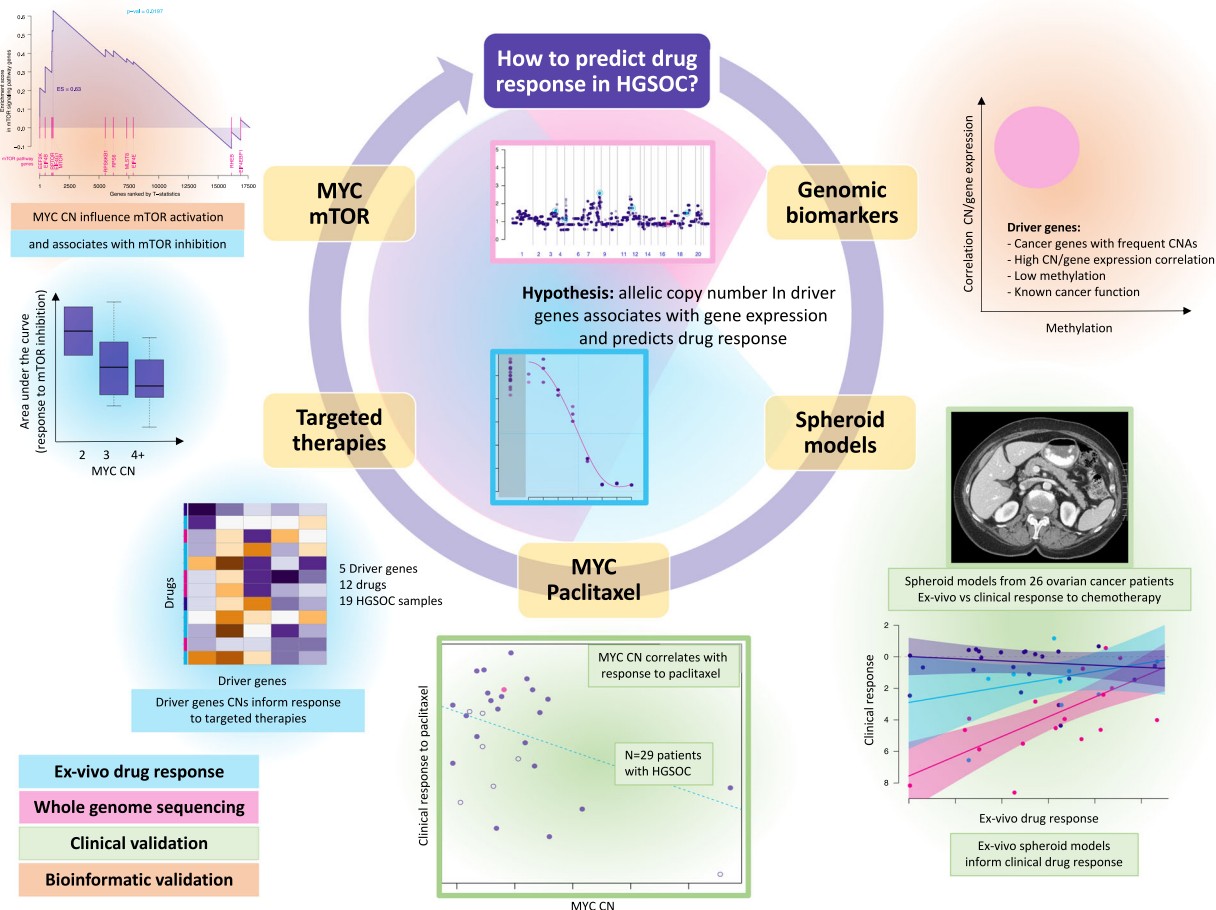

**Fig. 1** | Diagram summarising the research question, hypothesis and main results.

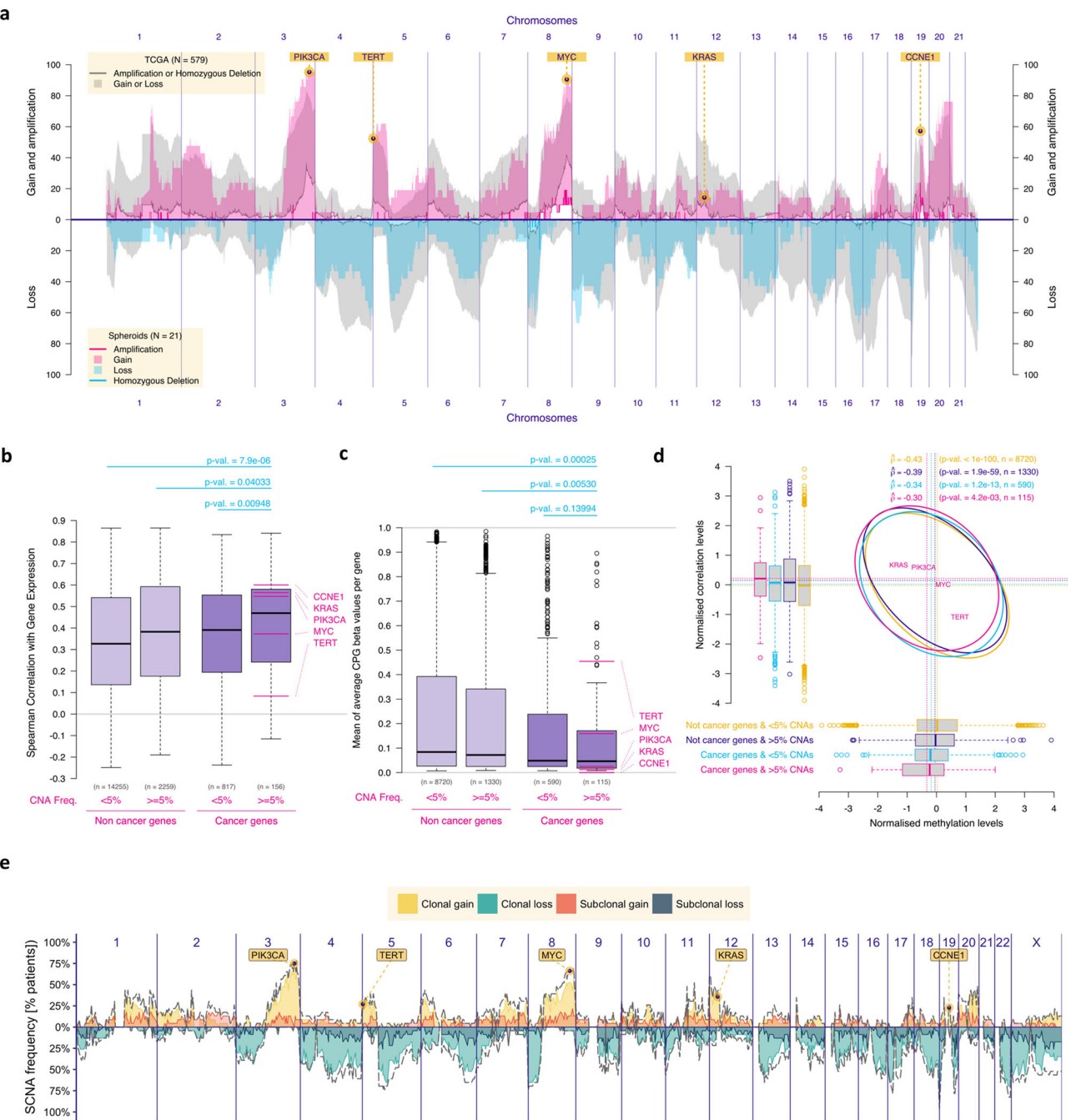

**Fig. 2 | Genomic analysis of single and multi-regional HGSOC cohorts defined clonal SCNA driver genes. a** Plot showing the prevalence of chromosomal alterations across the genome in both the TCGA cohort (*n* = 579) and in the HGSOC spheroid samples from the OV04 cohort (*n* = 21). For the spheroid cohort, gain was defined as 3 or 4 chromosomal adjusted copies and amplifications as ≥5 adjusted copies. **b** Boxplots showing the Spearman's correlation scores between gene expression and respective chromosomal copy number for each gene, split between cancer vs non-cancer genes and prevalent (>5% SCNAs in HGSOC) vs non-prevalent genes. Driver genes (in the far right; defined as 'cancer genes' that have SCNA alteration frequency in ≥5% of the samples) had the highest positive correlation scores. Numbers above the boxplots correspond to the p-values obtained with two-sided Mann-Whitney-Wilcoxon tests. **c** Boxplots showing methylation levels (beta-values) for all genes, split as cancer and non-cancer genes and as prevalent (>5% SCNAs in HGSOC) and non-prevalent genes. Prevalent non-cancer genes were significantly more methylated than prevalent cancer genes (two-sided Mann-Whitney-Wilcoxon's *p*-value: 0.005). **d** 95% prediction confidence ellipses displaying the estimated Pearson's correlation between the methylation levels (x-axis,

normalized) and the correlation between chromosomal copy number and gene expression (y-axis, normalized) for four groups of genes defined as combinations of cancer and non-cancer genes and as prevalent and non-prevalent genes. The boxplots respectively report the same information as the ones of panels **b** (vertical boxplots) and **c** (horizontal boxplots) on the normalized scale. For each group, Pearson's correlation estimate, *p*-value of the two-sided test of association between paired samples and number of genes are indicated (top right); for panels **b**, **c** and **d**, *n* = 371 independent TCGA samples, inference without multiplicity correction, and for all boxplots, the central box was defined by the quantiles 0.25, 0.5 and 0.75 of the data, and the wiskers as 1.5 times of the interquartile range. **e** Frequency of somatic clonal and subclonal copy number alterations across the genome of 72 tumour regions from 28 HGSOC primary tumours. Gains and losses were classified relative to ploidy (Supplementary Fig. S1b shows the genomic distribution of the frequency of somatic copy number alterations across 127 tumour regions of primary tumours and metastases from 30 HGSOC patients). The dotted line corresponds to the total number of gains and losses (clonal and subclonal).

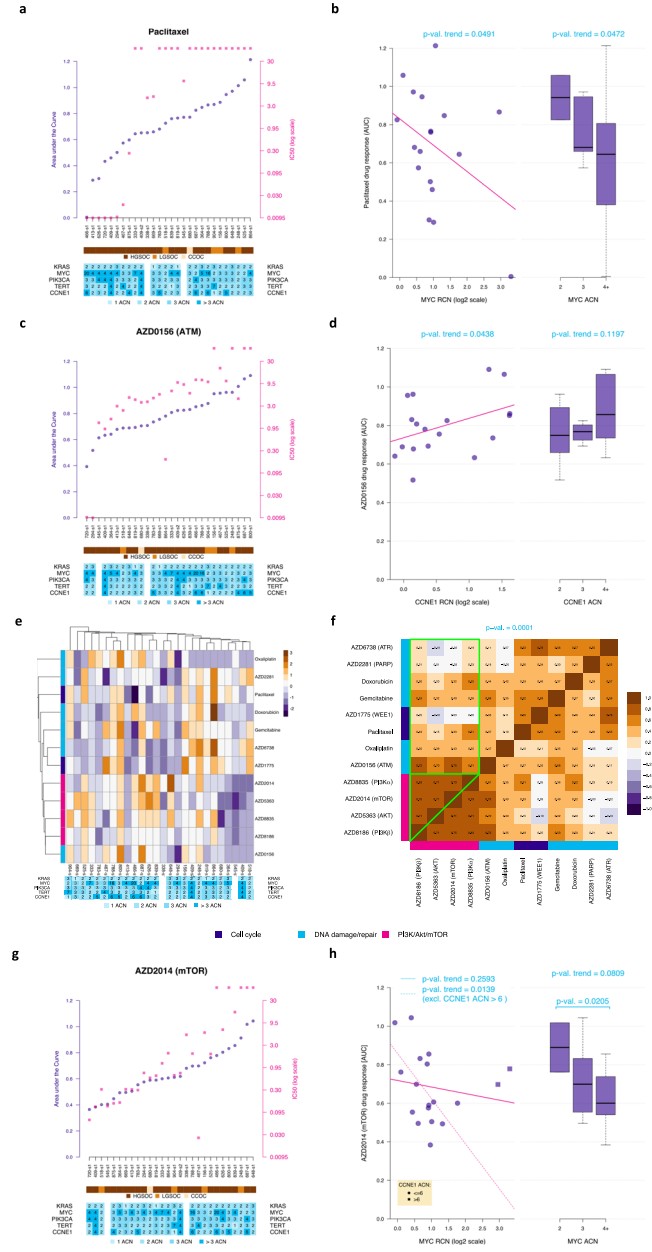

**Fig. 3 | Copy number of clonal SCNA driver genes informs drug response.**
**a** Scatterplot showing paclitaxel response measured by AUC (purple; left y-axis) and IC50 (pink; right y-axis) for all samples (n = 28) ordered by AUC levels. The lower bars show, for each sample, the histological diagnosis (HGSOC – high-grade serous; LGSOC – low-grade serous; CCOC – clear cell) and the normalised copy number for MYC, PIK3CA, KRAS, CCNE1 and TERT. Two of the HGSOC samples had failed sequencing data. IC50 dots above the scale of the figure represent samples where IC50 was not determined (viability was above 50% at the maximum dose). **b** Scatterplot and boxplots showing the associations between response to pacli- taxel in vitro (measured by AUC) and MYC relative copy-number (RCN; left plot) and normalised absolute copy-number (3-level ACN; defined by the absolute numbers normalised for a diploid genome, to allow comparisons, observed in n = 18 independent HGSOC samples; right plot) One-sided test p-values corre- sponding to the presence of a trend (linear model Wald t-test on the left and Jonckheere-Terpstra test on the right) are indicated. Regression effect size for the correlation between MYC RCN and response to paclitaxel is −0.4. **c** Scatterplot showing AZD0156 (p-ATM inhibitor) response measured in each sample following the same format as in panel a. **d** Association between CCNE1 RCN and ACN and AZD0156 in-vitro response (as in Fig. 3b); number of independent samples and statistical test identical to panel b; regression effect size of 0.4. **e** Two-dimensional hierarchical clustering of Z-scores for response of spheroids to each drug as observed in n = 26 samples (after exclusion of the 2 samples with extreme varia- bility). Adjusted copy number data for each spheroid is tabulated below the heat- map. Rows are colour-coded by target pathway for each drug. **f** Heatmap showing the Spearman's Rho correlation between in vitro response to different drugs observed in the HGSOC samples. Response to drugs affecting the PI3K pathway (pink) tend to be similar (high correlation values; green triangle), whilst the cor- relation between response to PI3K drugs and other drugs is lower (green square). The p-value corresponding to the non-parametric bootstrap test comparing these two sets of correlations is indicated. **g** Scatterplot showing AZD2014 (dual mTOR inhibitor) response measured in all samples following the format in Fig. 3a. **h** Association between MYC RCN and ACN and AZD2014 in-vitro response (as in panel 2b) when including (solid line) or excluding (dashed line) samples with high (>6) CCNE1 ACN; number of independent samples and statistical test identical to panel b; regression effect size of −0.2. For all boxplots in b, d and h, the central box was defined by the quantiles 0.25, 0.5 and 0.75 of the data, and the maximum whisker size equals 1.5 times of the interquartile range.

spheroid samples and matched primary tumours showed a strong correlation (Supplementary Figure S2c). We identified the adjusted copy number for each clonal driver gene (which is relative to and adjusted for ploidy) in each patient. Gains were defined as 3 or 4 copies and amplifications as 5 or more copies (Supplementary Fig. S4a).

We then assessed the performance of in vitro drug response from spheroids, by comparing it with the clinical response in patients from whom spheroids were derived and found that correlation between in vitro and clinical drug response was strongest for first-line che- motherapy (generally entailing carboplatin and paclitaxel) (Supple- mentary Fig. S5a–g).

Considering the number of samples we had and in order to minimise the risk of type I errors and maximise our power, we decided to focus on a set of predefined associations. Therefore, we tested for associations between clonal SCNA events and drug response associations that had been identified from previous screens. Previous genome-wide siRNA screens identified MYC as a paclitaxel sensitizer[33]. A higher number of MYC copies was asso- ciated with better response to paclitaxel in our spheroid models (p- value<0.05; Fig. 3a, b). Topoisomerase inhibition was shown to be synthetically lethal in the context of KRAS mutant colorectal cell lines[34]. Gain of KRAS in our HGSOC spheroids was associated with better response to doxorubicin (Supplementary Fig.S4b and c; p- value = 0.04). We then investigated whether these findings could be observed in a clinical cohort of 64 patients (Supplementary Table 2; Supplementary Fig. S2a) which showed a correlation between rela- tive copy-number of MYC and magnitude of CA125 change in

to mirror the distribution of clonal SCNAs (Fig. 2e and Supplementary Fig. S1b) as previously described in other tumours[6]. The distribution of focal clonal changes (e.g. gains or amplifications in 3q and 8q; Fig. 2e) most frequently encompassed the HGSOC drivers PIK3CA and MYC, as well as TERT, KRAS and CCNE1[9].

## Copy number of clonal SCNA driver genes informs drug response

To test associations between clonal SCNA drivers and drug response, we derived primary HGSOC spheroids from human ascites obtained at different points of the treatment (Methods; Supplementary Table 1; Supplementary Fig. S2a and S3a–u). We performed targeted deep sequencing and shallow whole genome sequencing (sWGS) on 26 primary ovarian carcinoma spheroids of which 21 were HGSOC and had adequate numbers of vials for functional analysis (Methods). Mutations in known drivers (with the exception of TP53) were uncommon (Supplementary Fig. S2b). The frequency distribution of SCNAs across the genome was similar between our spheroid samples and the TCGA HGSOC cohort (Fig. 2a). Equally the comparison between genomic profiles of six

response to carboplatin/paclitaxel (*p*-value = 0.005; Supplementary Fig. S4d). This suggests that *MYC* amplification could be a predictive clinical biomarker of response to those drugs.

To further test the hypothesis that clonal SCNAs may determine response to molecularly targeted agents, we assessed drugs targeting the PI3K pathway, cell cycle and DNA repair mechanisms, specifically testing correlations with the copy number of the clonal SCNA drivers we had identified (*MYC, PIK3CA, KRAS, CCNE1* and *TERT*).

*CCNE1* amplification is frequently present in tumours with competent homologous recombination pathways and has been associated with platinum-based chemotherapy resistance[35]. In the context of HRD, where co-existing *CCNE1* amplification is uncommon[36], the ATM pathway is frequently activated. Our spheroid model showed that *CCNE1* amplification was associated with platinum-resistance (Supplementary Fig.S4e) and that lower *CCNE1* copy number spheroids respond better to ATM inhibition (AZD0156; Fig. 3c, d and Supplementary Fig. S4e; *p*-value = 0.04).

Our data was also consistent with previous evidence suggesting cross-talk between mitogenic Ras/MAPK and survival PI3K/AKT pathways[37], as *KRAS* copy number was positively correlated with response to AKT inhibition (AZD5363; Supplementary Figure S4e). Additionally, signalling via the mTOR pathway has been shown to regulate translational and post-translational telomerase activity[38] and *TERT*-amplified HGSOC spheroids were susceptible to inhibition of PIK3CA, AKT and mTOR (Supplementary Fig. S4e and S4f).

Finally, despite the wide range of inter-patient response to the different drugs tested, some spheroid samples responded similarly to all the inhibitors targeting the PI3K/mTOR pathway (Fig. 3e). Pairwise comparison between responses to all drugs showed the highest correlation between inhibitors of the PI3K/mTOR pathway. This suggests that targeting any of the nodal points from this survival pathway has a similar effect on viability of HGSOC spheroids (*p*-value = 0.0001; Fig. 3f). Most strikingly, HGSOC spheroids with *MYC*-amplification were also more sensitive to dual m-TORC1/2 inhibition than tumour samples with normal MYC copy number (Fig. 3g, h, Welch *p*-value = 0.02; Jonckheere-Terpstra *p*-value = 0.08).

Considering the large number of possible combinations between gene copy-number and drug response and the limited sample size, we pre-selected five of the associations above to infer how driver SCNAs correlate to drug response (Supplementary Fig. S6a). We further performed a secondary analysis of 14 presumed associations out of 60 possible combinations (Supplementary Fig. S6b). Our results showed that, in our cohort, the associations between driver SCNA and drug response followed the expected pattern, supporting our hypothesis that clonal copy number changes in gene drivers predict drug response.

### MYC-amplified HGSOCs are associated with somatic copy number aberrations in genes from the NF1/KRAS and PI3K/AKT/mTOR pathways and activation of the mTOR pathway

To test the association between *MYC* amplification and dual m-TORC1/2 inhibition, we asked whether expression of mTOR-related genes was correlated with *MYC* expression using the TCGA HGSOC cohort. The expression of mTOR pathway genes were strongly correlated to *MYC* expression (*p*-value = 0.02) with negative correlations for the inhibitors of the pathway *RHEB* and *EIF4EBP1* (Fig. 4a, b). We hypothesised that activation of the mTOR pathway is required for anti-apoptotic survival mechanisms in the context of high *MYC*[39]. Indeed, the prevalence of SCNA affecting genes from the mTOR and RAS survival pathways were significantly higher in HGSOC cases with increased MYC copy numbers (Fig. 4c, *p*-value<0.0001). More specifically, amplification of *PIK3CA, IGFR1, GAB2, PTK6, KRAS* and *AKT1/2/3*, as well as deletion of *NF1* and *PTEN* (which lead to activation of the RAS and PI3K pathways, respectively) were significantly associated with *MYC* copy number (Fig. 4d). These data suggest that co-occurrence of

SCNAs affecting these genes and *MYC* amplification is an evolutionary requirement.

### Co-occurrence of MYC and survival PI3K/RAS SCNAs in other chromosomally unstable tumours

In order to test whether our observations in HGSOC of co-occurrence of *MYC* amplification with compensatory survival SCNAs could also be observed in other cancers driven by chromosomal instability[40], we compared the genomic landscape of HGSOC (Fig. 4c) to the profile of SCNAs in *TP53*-mutant triple-negative breast tumours (TCGA cohort, *n* = 261, Fig. 5a; Metabric cohort, *n* = 177, Fig. 5b) and squamous lung cancer (TCGA cohort, 501, Fig. 5c). We detected several genes that are frequently amplified or deleted across all these tumours, in the context of *MYC* amplification or gain (highlighted in yellow across Figs. 4c, 5a–c; e.g. *CCNE1, MCL1, PIK3CA, AKT1/2/3, RPTOR*). More importantly, we found a significant increase in the SCNAs encompassing genes from the PI3K and RAS pathways, compared to other cancer genes, in the context of *MYC* amplification or gain in the TCGA cohorts (TCGA breast cohort *p*-value = 0.0001; TCGA lung cohort *p*-value<0.0001; Fig. 5a, c). In *TP53*-mutant triple-negative breast tumours from the Metabric cohort, the overall rate of SCNAs was lower. Despite the significant overlap between SCNAs affecting tumours in these three distinct organs, there was a small number of genes that were specifically amplified or deleted in certain tumours (e.g. *ERBB2* amplification in *TP53*-mutant breast cancers and *CRKL* amplification or deletion of *LRP1B* or *CDKN2B* in squamous lung cancer). Interestingly, SCNAs in the majority of these "private" genes are also known to lead to the activation of the same survival mechanisms, suggesting that they are preserved across different tumours.

## Discussion

The extreme genomic complexity of HGSOC has hindered stratification for testing of new molecularly targeted therapies in the clinic. Previous SCNA analyses with GISTIC identified putative HGSOC drivers but these have only had limited functional validation in clinical samples[31]. We, therefore, tested whether CIN-induced SCNAs affecting expression of HGSOC driver genes are positively selected during tumour progression and show here that the correlation between SCNA and gene expression is higher in frequently amplified HGSOC cancer genes than in non-cancer genes from the same amplicons. Equally, promoters of cancer genes are less methylated than the promoters of non-cancer genes. Overall, these data suggest that aberrantly high expression from amplified HGSOC driver genes is crucial for initiation, cellular survival and/or progression of the disease. This concept is supported by a previous in vivo screen for tumorigenic SCNA in HGSOC that identified *GAB2* amplification as a driver of PI3K pathway activation and was associated with sensitivity to PI3K inhibition in cell lines[41].

Precision medicine approaches for women with HGSOC should target therapies to early, clonal drivers, or to subclonal drivers that are present in a substantial proportion of tumour cells[42]. The use of shallow whole genome sequencing is a cost effective way to profile multiple sites. Here, we provide comprehensive evidence that the chromosomal copy number of HGSOC driver genes affected by clonal SCNAs correlates with the response to specific chemotherapeutic and targeted agents both in vitro and in patients. We showed that *MYC* and *PIK3CA* gain and amplification were amongst the most frequently clonal alterations in HGSOC progression. Deregulated MYC promotes further chromosomal instability by affecting multiple aspects of mitotic chromosome segregation[43,44]. As increased c-MYC levels have been associated with platinum resistance[45], it is plausible that clinical responses to combined carboplatin and paclitaxel in MYC-amplified tumours are due to the paclitaxel effect.

Our data showed that *MYC* amplification or gain co-occur with SCNAs in PI3K genes which induce activation of the mTOR survival

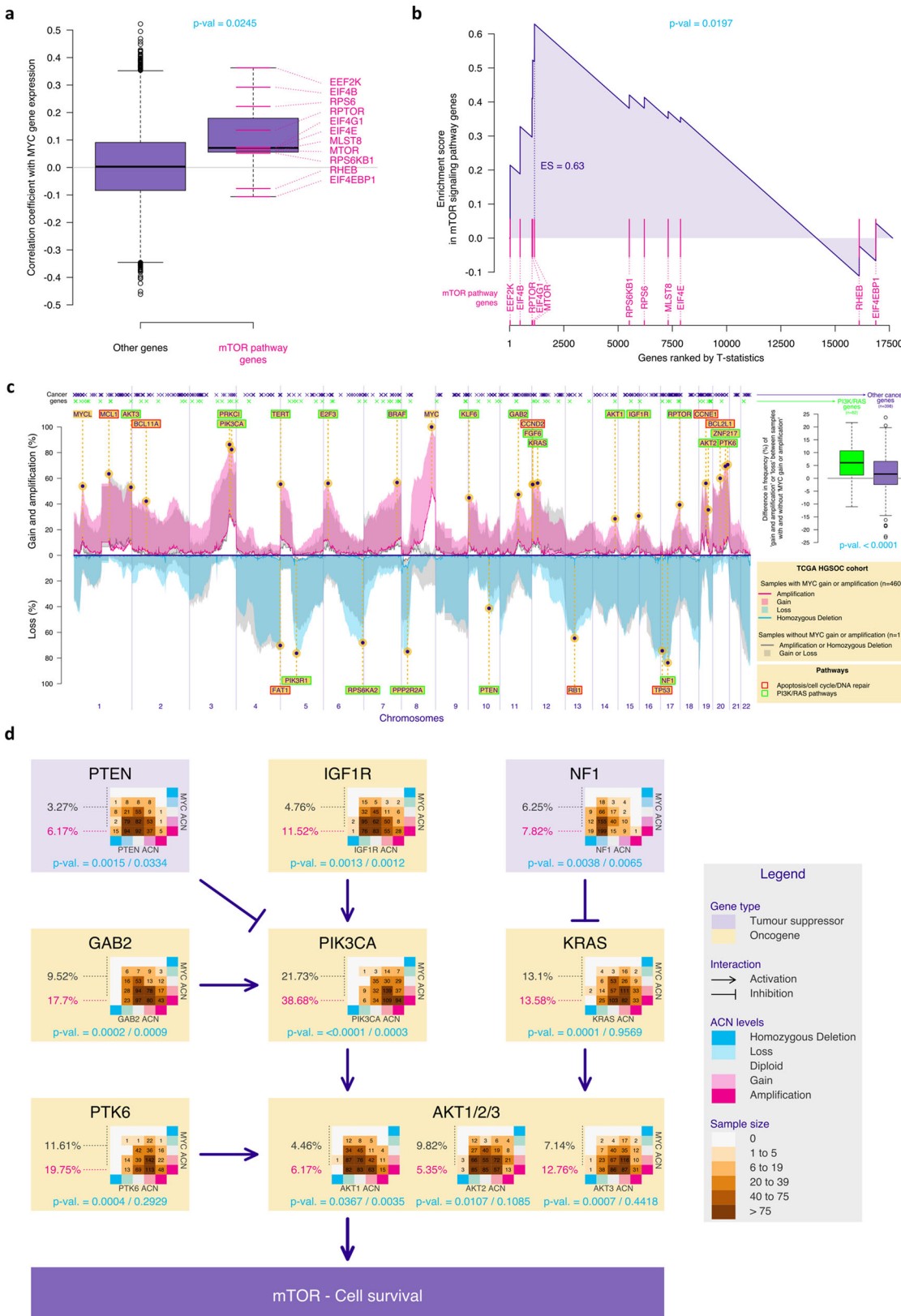

mechanisms and that mTOR inhibition was most effective in the context of *MYC*-amplified HGSOC samples. Myc-regulated protein synthesis is modulated by the mTOR-dependent phosphorylation of eukaryotic translation initiation factor 4E binding protein-1 (4EBP1), which is required for cancer cell survival in Myc-dependent tumours[46]. Inhibition of mTOR has been shown to be synthetically lethal in MYC-

driven haematological tumours[46]. Intriguingly, a complete response was previously observed in a patient with recurrent HGSOC during therapy with paclitaxel and the dual mTORC1/2 inhibitor vistusertib (AZD2014) in tumour lesions with a mutation in *MYC*[27]. The non-synonymous mutation was within the Myc homology-box 2 (MBII) domain which is a highly conserved region across species and myc

**Fig. 4 | MYC-amplified HGSOCs are associated with SCNAs in genes from the NF1/KRAS and PI3K/AKT/mTOR pathways and activation of the mTOR pathway. a** Boxplots showing the Pearson's correlation coefficient between the gene expression of all genes and the one of MYC, for genes belonging ($n = 11$) or not belonging ($n = 17638$) to the mTOR signalling pathway, based on BioPlanet annotation. The latter group showed, on average, higher correlation estimates compared to the other group (two-sided Mann-Whitney-Wilcoxon test). **b** Gene set enrichment analysis (GSEA) enrichment scores showing enrichment of mTOR signalling pathway genes in MYC-high tumours. The vertical pink lines represent the projection of individual genes from the mTOR pathway onto the gene list ranked by MYC expression level. The curve in blue corresponds to the calculation of the enrichment score (ES) following a standard two-sided GSEA. The more the blue ES curve is shifted to the upper left of the graph, the more the gene set is enriched in MYC-high genes. The ES score, the normalised ES score (NES) and p-value are also shown in the plot. **c** Frequency plot showing the distribution of chromosomal amplifications/homozygous losses (solid lines) or gains/heterozygous losses (shaded areas) across the genome in both MYC-amplified/gain (pink for amplifications/gains and blue for losses) and MYC diploid HGSOC (gray) in the HGSOC TCGA cohort. The location of a list of functional cancer genes selected in ref. 49 is indicated on top. Cancer genes are colour-coded in green if they belong to the PI3K or RAS pathways based on the Reactome definition. The boxplots (right panel) show, for both PI3K/RAS and other cancer genes, the difference between the frequency of cancer genes SCNAs in tumours with and without MYC amplification or gain. The p-value of the one-sided permutation test of equality of means is indicated. For all boxplots (in a and c), the central box was defined by the quantiles 0.25, 0.5 and 0.75 of the data, and the maximum whisker size equals 1.5 times of the interquartile range. **d** Diagram showing HGSOC drivers that impact the PI3K pathway and the prevalence of SCNAs across MYC allelic copy numbers (table). For each gene, the p-values corresponding two tests of association between both sets of absolute copy number are indicated (Chi-square test on the left and generalized Cochran-Mantel-Haenszel test for ordered factors on the right) are indicated in turquoise.

---

isoforms[47,48], suggesting increased dependence on *MYC* activation in this patient. Overexpression of c-myc induces apoptosis, which was previously shown to be reverted by overexpression of IGF-1, most likely through activation of the PIK3CA/AKT/mTOR survival pathway[39].

By demonstrating that other CIN-driven malignancies (e.g. triple negative breast cancer and squamous non-small-cell lung carcinoma) have similar genomic landscapes to HGSOC, our data suggest that survival mechanisms active in HGSOC are present across other CIN-driven tumours. This is corroborated by a recent study analysing the functional evolutionary dependencies in cancer. In this study, co-alteration of *PIK3CA* and the nuclear factor *NFE2L2* (whose transcription is partly regulated by *MYC*) was a synergistic evolutionary trajectory in squamous cell carcinomas[49].

Without focusing on the validation of individual clinical biomarkers, we have shown that rigorous identification of early/clonal driver SCNAs and understanding how the gene expression changes in important driver genes impacted by the SCNA affect cell fitness, proliferation and survival, may provide important clues to understand drug response. Low-depth whole-genome sequencing is becoming increasingly affordable and available. This work supports the routine use of these genomic tools and critically the importance of multi-region sampling to infer SCNA clonality for precision medicine trials in HGSOC and other CIN-driven malignancies.

## Methods

### Clinical samples and data
The patient samples were obtained from the CTCR-OV04 study, which is a prospective non-interventional cohort study approved by the local research ethics committee at Addenbrooke's Hospital, Cambridge, UK, (REC reference numbers: 07/Q0106/63; and NRES Committee East of England - Cambridge Central 03/018). OV04 study protocols are available at request. Clinical decisions were made by a clinical Multidisciplinary Team (MDT) and researchers were not directly involved. Written informed consent was obtained from all patients, and blood samples were collected before and after initiation of treatment with surgery or chemotherapeutic agents.

### Cohort 1
We obtained spheroids with primary tumour cells from ascites of twenty-eight patients with ovarian carcinoma that were recruited as part of the OV04 study in Cambridge University Hospitals. Twenty-three HGSOC samples from twenty-two patients were obtained (there were two matched samples from the same patient obtained before and after administration of a chemotherapeutic regimen).

### Cohort 2
Solid tumour samples were collected from 64 patients in Cambridge University Hospitals as part of the OV04 study (six of the tumour samples were taken from the same patients that also provided spheroid samples). Samples were handled on ice and processed as soon as possible after surgery. Tumour tissues were fixed for 24 h in 10% neutral buffered formalin (NBF) before being transferred to 70% ethanol and embedded in paraffin.

### Clinical data
Clinical data collected included dates of sample collection, surgery, treatment and therapy dates (where applicable), date of death and serial serum CA125-levels from diagnosis. CT reports were obtained and classified by Consultant Radiologists into categories which included progressive disease, stable disease, partial response or complete response. A summary of the clinical cohort is presented in Supplementary Table 1 and all the plots summarising clinical data are presented in Supplementary Fig.S2a and S3a–u.

### Spheroid isolation
Ascitic fluid was collected from patients with the volume ranging between 100–2000 mL volume. The fluid was centrifuged at 800 *g* for 5 min and the majority of the supernatant was removed. Blood clots were removed using a butter muslin cloth and the remaining sample was passed through a 40 μm filter. Spheroids were then recovered after a 10 ml wash with PBS and centrifugation at 400 *g* for 5 min. Next the spheroid fraction was divided in two. One portion of the cell pellet was utilised for DNA extraction. The other portion of the cells were resuspended in filtrated acellular ascitic supernatant and 8% DMSO, transferred to freezing vials and kept in liquid nitrogen. Cells were thawed and kept in media at 37 °C for 12 h before drug screening was performed. Purity of spheroids was both assessed semi-quantitatively through a pathologist review and/or quantitatively inferred from the TP53 allele frequency.

### Ex-vivo drug response/Spheroid assays
An 8-point half-log dilution series of each compound was dispensed into 384 well plates using an Echo® 550 acoustic liquid handler instrument (Labcyte) and kept at −20 °C until used. Prior to use plates were span down. 50 μl of spheroid suspension were added per well using a Multidrop™ Combi Reagent Dispenser (Thermo-Fisher). For each sample, the same vial was used to load both control and test wells, to control for spheroid size and cell density. Following 5 days of drug incubation, a cell viability assay using 30 μl of CellTiter-Glo® (Promega) was performed. We performed technical triplicates. Due to limited sample availability, biological replicates were only performed in selected samples.

### DNA extraction from FFPE samples
Multiple sections at 10 μm thickness were cut for each FFPE sample depending on tissue size and tumour cellularity. Tumour areas were

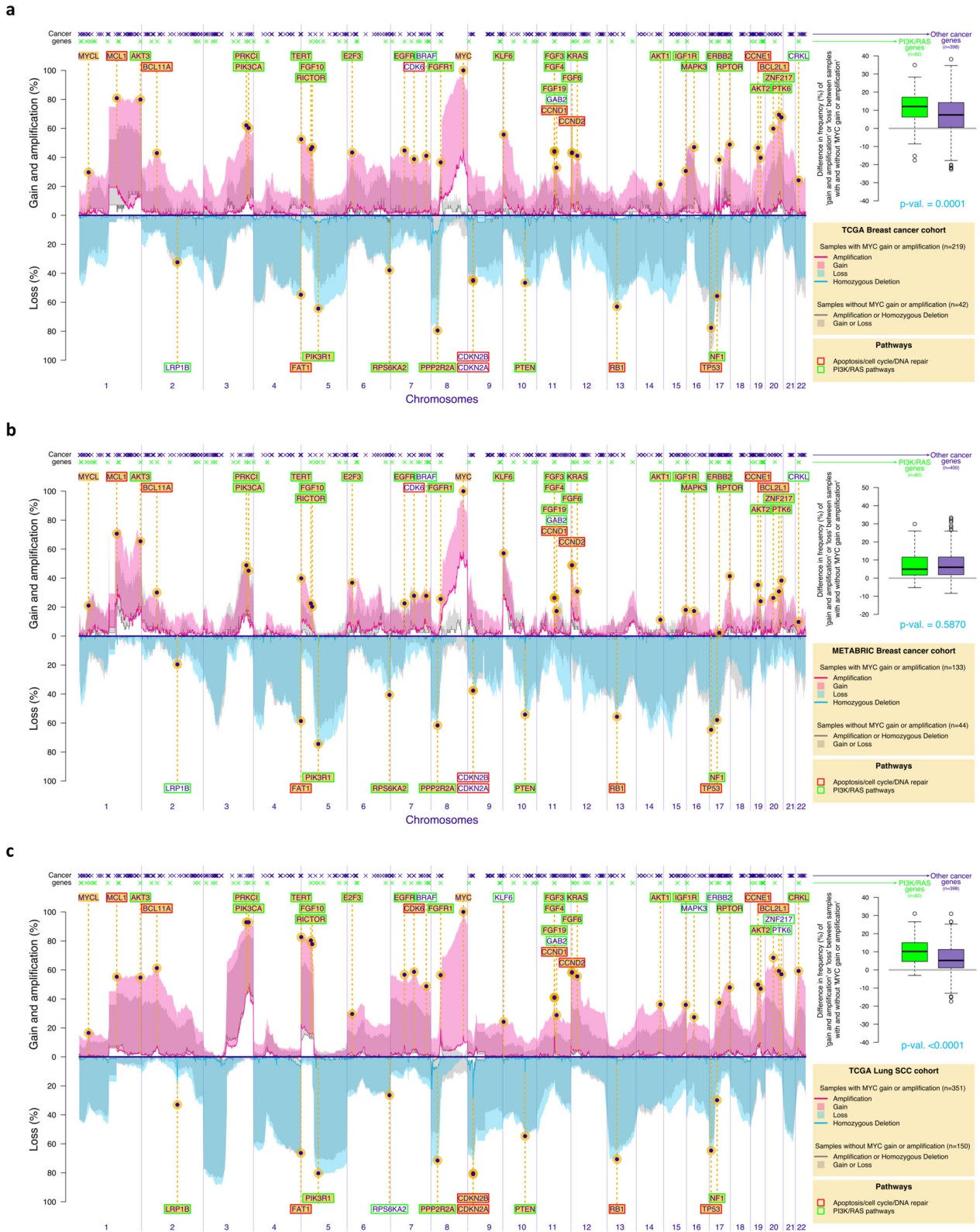

marked by a pathologist on separate Haematoxylin and Eosin (H&E) stained sections to guide microdissection for DNA extraction. Tumour areas from unstained tissue sections were scraped off and dewaxed in 1 ml of xylene, followed by washing in 100% ethanol. After residual ethanol was evaporated (10 min at 37 °C), DNA extraction was performed using the AllPrep DNA/RNA FFPE Kit (Qiagen). DNA was eluted

in 40 μl kit ATE elution buffer and quantified using Qubit quantification (Thermofisher, Q32851).

### Genomic profiling

DNA extraction was performed using the DNeasy Blood & Tissue Kit (QIAGEN) according to manufacturer's instructions. DNA samples

**Fig. 5 | Co-existence of MYC amplification and SCNAs from the PI3K and RAS pathways in lung squamous and triple-negative p53-mutant breast cancers.** Frequency plots showing the distribution of chromosomal amplifications/homozygous losses (continuous line) or gains/heterozygous losses (shade) across the genome in both MYC-amplified/gain (pink for amplifications/gains and blue for losses) and MYC diploid tumours (gray) in the Breast TCGA cohort (**a** triple-negative invasive ductal p53-mutant tumours only), Breast Metabric cohort (**b** triple-negative invasive ductal p53-mutant tumours only) and Lung Squamous TCGA cohort **c**. The location of a list of functional cancer genes selected in ref. 49 is indicated on top. Cancer genes are colour-coded in green if they belong to the PI3K or RAS pathways based on the Reactome definition. The boxplots (right panel) show, for both PI3K/RAS and other cancer genes, the difference between the frequency of cancer genes SCNAs in tumours with and without MYC amplification or gain. The *p*-value of the one-sided permutation test of equality of means is indicated. A list of known driver genes is also presented across all plots – the genes are highlighted in yellow if they are recognised GISTIC drivers in each specific tumour. For all boxplots (**a**, **b** and **c**), the central box was defined by the quantiles 0.25, 0.5 and 0.75 of the data, and the maximum whisker size equals 1.5 times of the interquartile range (inference without multiplicity correction).

were diluted to 75 ng in 15 µl of PCR certified water for subsequent shearing by sonication using the LE220-plus Focused-Ultrasonicator (Covaris) for 120 s at RT (duty factor 30%; Peak incident power 180 W; 50 cycles per burst) with a target of 200–250 bp. Library preparation was subsequently carried out using the SMARTer Thruplex DNA-Seq kit (Takara) with each sample undergoing 7 PCR cycles for library amplification and sample indexing. sWGS libraries were cleaned using AMPure beads, according to the manufacturer's recommendations, and eluted in 20 µl TE buffer. Quality and quantity of sWGS libraries were assessed using a D5000 genomic DNA ScreenTape (Agilent, 5067-5588) on the 4200 TapeStation System (Agilent, G2991AA).

Libraries were pooled and sequenced using the Paired-End 50 mode and S1 flowcell on the NovaSeq. Genomes were aligned to the GRCh37 reference genome and relative copy number data was obtained using the qDNAseq package(24).

### Tagged-amplicon sequencing (TAm-Seq)
Targeted sequencing of 697 amplicons in BRCA1,BRCA2,RAD51C,RAD51B,RAD51D,BARD1, BRIP1,PALB1,FANCM,TP53, PTEN, EGFR, CDK12, hot spots: BRAF, KRAS, PIK3CA, POLE, CTNNB1, NRAS genes was performed using Fluidigm Access Array 48.48 platform as described previously[50,51]. No preamplification step was applied. 50 ng of DNA was used as the library input. All libraries were pooled and quantify using Agilent 4200 Tapestation according to the supplier's recommendations (D1000). Libraries for all samples were prepped, sequenced and analysed in duplicates. Prepped libraries were sequenced using PE-150 mode on Ilumina HiSeq4000 aiming for 500–1000x depth. For curation of TAM-seq data, we estimated the functionality of identified gene variants (single nucleotide variants and indels) using the MTBP clinical decision support system. The functional relevance of a variant is classified using a 3-tier system: putative functional, putative neutral and variant of unknown significance. Furthermore, putative functional variants were annotated using multiple sources of evidence to rank them into evidence type A (curated effect across queried knowledge bases), evidence type B (evaluated according to bona fide assumption), and evidence type C (evaluated by computational-derived metrics). We colour coded the identified MTBP annotations in order to build a heat map of all gene variants[52].

### Clonality of SCNA events in multiregional whole genomes from primary tumours and metastases in HGSOS
We analysed the clonality of copy number events in a cohort of 30 high-grade serous ovarian cancer patients, using multiregional whole-genome sequencing[32,53]. Allele-specific copy number and ploidy estimates for 127 regions were derived using the tool for allele-specific and clone-specific copy number estimation, Remixt[32,53,54]. For each patient, we defined a tumour consensus copy number profile and masked low-mappability and sequencing-inaccessible regions. We next calculated non-allele-specific raw copy numbers using the major and minor raw copy number values normalized by ploidy using Eq. (1).

$$raw\_total\_CN = \sum \log_2\left(\frac{raw\ minor + raw\ major}{ploidy}\right) \quad (1)$$

Copy number states, gains and losses, were classified based on the raw non-allele-specific copy number (*raw_total_CN*) using the thresholds described in Eqs. (2) and (3):

$$Gain : raw\_total\_CN > \log_2\left(\frac{2.5}{2}\right) \quad (2)$$

$$Loss : raw\_total\_CN < \log_2\left(\frac{1.5}{2}\right) \quad (3)$$

Clonal gain was defined as a gain event detected in all samples per patient, while subclonal gain events were present in a subset of the samples. Clonal loss is defined as an LOH event or loss event relative to ploidy detected in all samples from a patient. Alternatively, the loss events present in a subset of patients were called subclonal.

### Statistics & reproducibility
All statistical tests were two sided, unless stated otherwise.

**Sample size.** The sample size was determined by the number of available spheroids at the time of analysis. An 'effect size' analysis was performed and determined that the available sample size was able to detect biologically meaningful associations between in-vitro and in-vivo drug response (slope coefficient) by means of one-sided tests (without multiplicity correction) with a probability/power greater than 0.8.

**Data exclusions.** There were a total of $n = 28$ spheroid ovarian carcinoma samples (23 of which were HGSOC; 2 of HGSOC samples had failed sequencing). For associations between clonal SCNAs and drug response, we excluded non-HGSOC spheroid samples ($n = 5$), samples with failed sequencing data ($n = 2$), samples with extreme variability in the in-vitro studies ($n = 2$) and, in statistical analyses requiring independence between observations, the second of the two samples of patient 409.

**Replication.** We have performed technical triplicates in the drug response assays and, in samples with enough biological material, biological duplicates. All attempts were successful.

**Randomization.** In the drug response assays, the edge wells were left empty to prevent edge effects. Analysis of the data didn't suggest the presence of a pattern induced by the mapping of the drug. Randomization was not felt necessary since both spheroids and drugs were dispensed by automated devices.

**Blinding.** The standard blinding is not applicable in this study considering retrospective clinical data. However, blinding to the experimental was kept during data collection and data analysis

**a/ Experimental design for the analysis of associations between drug response and adjusted copy number.** We selected 12 drugs, four of which were representative of conventional chemotherapeutic agents and eight new targeted inhibitors. The 8 new compounds were

selected based on their ability to impair DNA repair and the PI3K pathway, which are well known to be crucial for HGSOC survival.

We aimed to test the impact of clonal alterations in drug response. Considering our limited sample size, we initially tested the impact of multiple testing and sampling error prior to our experimental analysis using a simulation study that considered 20 patients, 12 drugs and 5–10 copy number changes. This showed that, when generating data under the null hypothesis (ie, assuming independence between drug response and gene copy number), absolute Pearson correlation estimates of all pairwise 'drug and gene' combinations would range, on average, between 0.0–0.6. Considering the large probability of observing spurious relationships between gene copy number and drug response, we decided to only focus on five robust associations based on the literature. They correspond to associations well established in previous screens (high MYC – response to Paclitaxel; high KRAS – response to Doxorubicin) or that could be predicted based on the known functional impact of the specific SCNAs (high CCNE1 – resistance to pATM inhibitor; high MYC – response to mTOR; high TERT - response to mTOR).

We also performed a secondary analysis using an extended this list of 14 gene-drug associations supported by previous studies (Details in Supplementary Material and Supplementary Fig. S6).

**b/ Test of equality of the location parameter between two populations.** In the Figs. 2b, c and 4a, we used Wilcoxon rank sum non-parametric tests (also known as Mann-Whitney-Wilcoxon tests) with continuity correction (*wilcox.test* function of the *stats* R package) to assess if the location shift between two populations is different from 0. Similar conclusions were obtained when considering Welch *t*-tests (*t.test* function of the *stats* R package) and median test (*median_test* function of the *coin* R package). Figure S1a repeats the analysis of Fig. 2b for different threshold values for prevalence of altered versus non-altered cancer genes over the range of interest (between 0 and 0.1).

In the Figs. 4c, 5a–c, we used one-sided permutation tests (considering R = 10'000 resampling of the TCGA or METABRIC patients) to assess if the mean of the differences (in %) of 'amplification or gain' or 'loss' (reverse scale) levels of tumours with or without MYC gain or amplification are different for cancer genes ref. [49] belonging or not to the Reactome PI3K or RAS pathways. Triple-negative breast tumours from TCGA cohort (Fig. 5a) were selected based on the Code 8500/3 from the International Classification of Diseases for Oncology, Third Edition ICD-O-3 Histology Code. Triple-negative breast tumours from Metabric cohort (Fig. 5b) were selected based on negativity for ER, PR and HER2 status.

**c/ Estimates, test and representation of the association between paired samples.** In the Fig. 2b, we estimated the Spearman's correlation coefficient (*cor* function of the *stats* R package) between normalised gene expression (obtained with the function *cpm* of the *edgeR* R Bioconductor package) and respective chromosomal copy number for each gene in the HGSOC TCGA database. Similar conclusions were obtained when considering Pearson's product moment correlation estimates.

In the Fig. 2d, we used the Pearson's product moment correlation coefficient test (function *cor.test* of the *stats* R package), to assess if the level of association between 2 variables is different from 0 (similar conclusions obtained when considering Spearman correlation tests). In the Fig. 2d, the covariance/correlation matrix and the vector of means were normalised (by considering the quantiles of a standard normal distribution corresponding to the observed probability point of each gene) since they are the sufficient statistics of the bivariate normal distribution. Representation of the level of association between the normalised variables of interest was then obtained by

displaying ellipses corresponding to the quantile 0.95 of the bivariate normal distributions of interest.

In the Fig. 3f, the association between the ex-vivo response to paired drugs was defined by means of Spearman rank-based correlation (*cor* function of the *stats* R package). We estimated (i) the level of association between ex-vivo response to each pair of drugs affecting the PI3K/Akt/mTOR pathway, (ii) the level of association considering all paired drugs, including one drug from this PI3K/Akt/mTOR group and one from another group, and compared their mean by means of a non-parametric bootstrap test considering 25000 samples (the corresponding *p*-value is indicated).

In the Fig. 4a, we estimated the Pearson's product moment correlation coefficient (function *cor* of the *stats* R package) between expression of MYC and expression of all the other genes from the HGSOC TCGA cohort.

In the Fig. S2c, we estimated the Spearman's correlation coefficient (*cor* function of the *stats* R package) between the relative copy number of the different genomic segments in tumour samples and in matched spheroid.

**d/ Response to treatment estimates.** in Fig. S3a–u, S4c and S4d, response to treatment was estimated using area under the curve (AUC) and IC50.

Drug response measures were standardised by dividing the original values by the median drug response observed in the control group of each drug and sample. These standardised drug response measures were then modelled as a function of the dose (on the log scale) by means of a 4th degree polynomial robust regression, fitted by means of the function *lmrob* of the R package *robustbase*. Drug response measures that obtained a robust weights smaller than 0.4 (out of a range which spreads from 0 for outliers to 1 for non-outliers) were considered as outliers. After excluding outliers, we modelled the standardised drug response measures as a function of the dose (on the log scale) by means of M-splines. AUC and IC50 were estimated using the I-splines (which correspond to the integrals of M-splines). The alternative use of a five-parameter log-logistic fit (*drm* function of the *drc* R package with function LL2.5) led to similar AUC and IC50 estimates.

**e/ Linear relationship between continuous variables with independent samples.** In the left plots from the Figs. 3b, d, h and S4c, we used linear regressions, fitted by means of the function *lm* of the *stats* R package, to model the relationship between drug response (AUC) and MYC relative copy number (on the log scale). The *p*-values of the one-sided Wald *t*-test corresponding to the slope parameter of each fit are indicated. Corresponding linear model effect sizes (corresponding to the parameters of standardised predictors) were estimated by means of the function *effectsize* function of the R package *effectsize*.

In the left plot of the Fig. 3h, the trend between drug response and MYC relative copy number was also fitted when excluding the two observations corresponding to samples with high CCNE1 amplification (ACN > 6; patients 466 and 788). The direction of the relationship was pre-specified[33].

We only considered the first sample for patient 409, since the linear model requires independent observations. Equally, the samples from patients 720 and 875 were excluded from these analyses due to extreme variability in the results demonstrated by the high standard deviations in the (standardised) drug responses of the control group (Figures S3a–u).

**f/ Trend test between an ordinal and a continuous variable.** In the right plots of Figs. 3b, d and h, we used one-sided Jonckheere-Terpstra tests (*jonckheere.test* function of the R package *clinfun*), to test for ordered differences in drug response between the 3 levels of the gene

adjusted copy number variable (2, 3 and 4+), assuming the same trend direction as in the plots on the left.

In Fig. 3h and S4c, we used Student's t-tests, (*t.test* function of the *stats* R package), to investigate a difference in means between two levels of KRAS and MYC ACN. The second sample of patient 409 and the samples of patients 720 and 875 were excluded from these analyses for the same reasons described above.

In Fig. S5a and S5b, we used an ordinal regression (*clm* function of the *ordinal* R package) as well as Jonckheere-Terpstra test (*jonckheere.test* function of the *clinfun* R package) to assess the level of association between the ordinal CT responses and different measures of changes in CA125 during a period of interest. Figure S5b focuses on the difference in CA125 on the log2 scale before and after treatment.

**g/ Linear relationship between continuous variables with dependent samples.** As patients typically have several lines of treatment and as their responses to treatment are likely not independent, linear mixed models with patients as random effects (to take the within-patient dependence into account) were used in analyses related to the clinical drug response. Models were fitted by means of restricted maximum likelihood (REML; *lme* function of the *nlme* R package).

In Fig. S5f, a heteroscedastic mixed linear model with [i] patients as random effects, [ii] AUC, line number (as a three-level factor: 1, 2 and 3+) and the interaction between AUC and line number as fixed effects and [iii] residuals as a power function of the duration of the chemotherapy cycle (to account for the fact that the variability increases with duration) was fitted. The Table on the bottom right of Fig. S5f shows the p-values obtained when comparing the slope of each line number (rows) to 0 (no relationship between ex-vivo and clinical response) and other line numbers (pairwise comparisons) using a multiplicity correction taking the dependence between these tests into account, as implemented in the function *ghlt* of the *multcomp* R package. Figure S5g considers the same analysis as Fig. S4f but uses data ignoring changes in CA125 occuring after surgery by either considering the changes in CA125 from one month post-surgery, or only pre-surgery changes in CA125, whichever the longest (number of days).

In Fig. S4d, random intercept models, considering patients as random effects and MYC relative copy number (on the log scale) as the fixed effect were fitted with or without controlling for tumour purity (obtained from TP53 mutant allele frequency). The p-values indicated correspond to the Wald t-test of the slope parameter obtained with or without tumour purity in the model.

**h/ Copy number frequency plots.** The copy number frequency plots in the Figs. 2a, 4c, 5a–c were obtained by computing the percentage of [i] homozygous loss, [ii] heterozygous loss, [iii] gain and [iv] amplification for the subset of samples of interest (all TCGA or spheroids samples in Fig. 2a; for samples showing MYC gain or amplification or neither of both in Figs. 4c, 5a–c). Amplification (homozygous loss) and combined gain and amplification (homozygous and heterozygous losses) were displayed for each gene and subset of interest. Considering a log2 relationship between relative and adjusted copy number, we used adjusted copy numbers of 0.5, 1.5, 2.5 and 4.5 (Fig. S4a) to define the limits of the subgroups in the spheroid cohort from Fig. 2a. in the Figs. 2a, 4c, 5a–c, we considered the absolute copy number available for the TCGA and METABRIC.

**i/ Differential gene expression analysis.** In the Gene Set Enrichment Analysis (Fig. 4b, we obtained a list of ranked genes according to the t-statistic corresponding to the change in mean gene expression intensities between HGSOC TCGA samples with MYC-low and MYC-high (lower and upper quartile of MYC expression respectively). We discarded 6522 out of 24165 genes with low counts across all samples (counts-per-million below 0.5 in more than 90% of the samples; from the *cpm* function of the *edgeR* R Bioconductor package). Differential expression analysis between the samples with low (smaller than the 1st quartile) and high (greater than the 3rd quartile) MYC gene expression levels was performed using the *voom* function of the *edgeR* R Bioconductor package (with default options).

**j/ Gene score enrichment analysis.** To assessed whether the up-regulated genes in MYC-high compared to MYC-low tumours were enriched for mTOR pathway genes, we obtained a list of genes belonging to the mTOR signalling pathway from the Bioplanet database (https://tripod.nih.gov/bioplanet/, the pathway "Mammalian target of rapamycin complex 1 (mTORC1)-mediated signalling") and ran a gene set enrichment analysis (*fgsea* function of the R Bioconductor package *fgsea*; Fig. 4b), using the mTOR gene list and the list of ranked genes of the differential gene expression analysis described above, to obtain the enrichment score estimate and corresponding p-value.

**k/ Clustering analyses.** In Fig. 3e, f and S4e, we used hierarchical clustering to group samples, drugs or genes. In the Figs. 3e and S4e, samples, drugs and genes were grouped according to a 'complete' hierarchical clustering based on correlation distances (*pheatmap* function of the *pheatmap* R package and *hclust* function of the *stats* R package).

In Fig. 3f, the drugs were grouped according to a 'complete' hierarchical clustering based on euclidian distances defined on the drug Spearman's correlation matrix (*hclust* function of the *stats* R package).

**l/ Tests of association of categorical or ordered factors.** In Fig. 4d, we present the association between the absolute copy number (ACN) of selected genes and the ACN of MYC in two different way. Firstly, we considered the ACN vectors as non-ordered factors and used the Chi-square tests (*chisq.test* function of the *stats* R package). Secondly, we considered them as ordered factors and used the generalized Cochran-Mantel-Haenszel tests of association of ordered factors (*CMHtest* function of the *vcdExtra* R package).

**m/ Tests of equality of continuous one-dimensional probability distributions.** In Figure S6a, we compare the empirical distribution functions of the p-values related to the analyses of Figs. 3b, d, h, S4c and S4f, to the uniform distribution by means of QQ-plots and formally test if these two distributions are different by means of Kolmogorov–Smirnov tests (ks.test function of the stats R package). As sensitivity analysis, we compare the Kolmogorov–Smirnov test p-value obtained when considering the original list of 5 drug - gene copy number associations (horizontal coloured line) with the p-values obtained a similar way when considering 1000 random sets of *drug response - gene copy number* associations (boxplot) generated by randomly selecting genes. Figures S6b consider the same analysis with an extended set of 14 drug - gene copy number associations.

### Reporting summary
Further information on research design is available in the Nature Research Reporting Summary linked to this article.

## Data availability
The shallow whole genome sequencing data and amplicon sequencing data for the 26 ovarian cancer patients analysed in this study have been deposited at the European Genome-Phenome Archive (hosted by the EBI) under accession number EGAD00001008716. Data is available under restricted access so that the use of human data is kept traceable. Please contact the corresponding author directly for access requests.

Sequencing data regarding retrospective studies (TCGA HGSOC, TCGA breast, TCGA lung and Metabric) are available for download through the Cbioportal platform [https://www.cbioportal.org] at the following links:

TCGA HGSOC [https://www.cbioportal.org/study/summary?id=ov_tcga_pan_can_atlas_2018]

TCGA breast [https://www.cbioportal.org/study/summary?id=brca_tcga_pan_can_atlas_2018]

TCGA lung [https://www.cbioportal.org/study/summary?id=nsclc_tcga_broad_2016]

METABRIC [https://www.cbioportal.org/study/summary?id=brca_metabric].

## Code availability

All data and code allowing reproducibility of all figures were shared in Zenodo [https://doi.org/10.5281/zenodo.6981371][55].

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

## Acknowledgements

We thank all the patients who participated in this study. We thank Prof. Samuel Aparicio for critical review of the manuscript and Dr. Suzanne Carreira for the identification of the MYC mutation in the patient that had complete response in the phase 1 trial described in (27). F.C.M. is funded by the Experimental Medicine Initiative from the University of Cambridge, the Academy of Medical Sciences (SGL016_1084), Cancer Research UK (C53876/A24267), the Addenbrooke's Charitable Trust (REF 13/17) and the Rosetrees and Stoneygate Trusts (A2854). This research was also supported by a pump-priming award from the Cancer Research UK Cambridge Centre Early detection Programme (CRUK grant ref: A25117). This research was supported by the NIHR Cambridge Biomedical Research Centre. The OVO4 study is supported by the CRUK Cambridge Cancer Centre and the Mark Foundation Institute for Integrated Cancer Medicine. We would like to acknowledge the support of The University of Cambridge, the National Institute for Health Research Cambridge, National Cancer Research Network, the Cambridge Experimental Cancer Medicine Centres, Hutchison Whampoa Limited and Cancer Research UK (CRUK grant numbers A22905 (JDB), A15601 (JDB), A25177 (CRUK Cancer Centre Cambridge)). C.S. is Royal Society Napier Research Professor (RSRP\R\210001. His team and work is supported by the Francis Crick Institute that receives its core funding from Cancer Research UK (CC2041), the UK Medical Research Council (CC2041), and the Wellcome Trust (CC2041). C.S. is funded by Cancer Research UK (TRACERx (C11496/A17786), PEACE (C416/A21999) and CRUK Cancer Immunotherapy Catalyst Network), Cancer Research UK Lung Cancer Centre of Excellence (C11496/A30025), the Rosetrees Trust, Butterfield and Stoneygate Trusts, NovoNordisk Foundation (ID16584), Royal Society Professorship Enhancement Award (RP/EA/ 180007), the National Institute for Health Research (NIHR) Biomedical Research Centre at University College London Hospitals, the CRUK-UCL Centre, Experimental Cancer Medicine Centre, the Breast Cancer Research Foundation (BCRF, USA) and The Mark Foundation for Cancer Research Aspire Award (Grant 21-029-ASP). His research is supported by a Stand Up To Cancer-LUNGevity-American Lung Association Lung Cancer Interception Dream Team Translational Research Grant (Grant Number: SU2C-AACR-DT23-17 to S.M. Dubinett and A.E. Spira). Stand Up To Cancer is a program of the Entertainment Industry Foundation. Research grants are administered by the American Association for Cancer Research, the Scientific Partner of SU2C. CS is in receipt of an ERC Advanced Grant (PROTEUS) from the European Research Council under the European Union's Horizon 2020 research and innovation programme (grant agreement No. 835297).

## Author contributions

F.C.M., C.S. and J.D.B. supervised the project. F.C.M and J.D.B. conceived and designed the project. C.S. conceived and supervised the evolutionary multi-regional analysis. F.C.M., R.C., C.S. and J.D.B. obtained funding for the study. F.C.M., D.S. and M.V. designed and performed the drug screens. F.C.M., A.A. and Ka.H. collected the clinical data. F.C.M. and D.L.C. analysed and interpreted molecular, experimental and clinical data and wrote the initial manuscript draft. D.L.C. led the bioinformatic and statistical analyses of the clinical, experimental and genomic data and designed the manuscript figures. I.S. performed the TCGA SCNA/gene-expression and gene-enrichment analysis and provided bioinformatic support throughout the work. C.M.S., M.V., A.P., J.H. collected and processed primary tumour and spheroid samples and prepared SGWS libraries. M.A. and A.W.M. performed the clonality analysis using the multi-regional HGSOC sequencing data. S.C., L.C., B.D. facilitated access to AZ drugs and drug screen facilities, and assisted with the analysis of results from the drug screen. G.F., H.B., Kr.H., J.L., P.B., R.C., B.B and M.J.L. assisted in data collection, patient recruitment, establishing avenues of enquiry related to patient clinical and molecular characteristics and manuscript editing. T.B.K.W., M.E., N.M., K.L. provided guidance on the bioinformatic analysis. S.S. provided processed multi-regional HGSOC sequencing data. F.C.M., D.L.C., B.B., S.S., I.M., C.C., G.E., C.S., J.D.B. reviewed the full analysis. All authors reviewed and approved the final submitted manuscript.

## Competing interests

The Experimental Medicine Initiative from the University of Cambridge that funded F.C.M. Clinical Lectureship is partly funded by Astrazeneca. C.C. is a member of the AstraZeneca (AZ) External Science Panel, and has research grants from Roche, Genentech, AZ, and Servier that are administered by the University of Cambridge and reports receiving speakers' bureau honoraria from Illumina. C.S. acknowledges grant support from Pfizer, AstraZeneca, Bristol Myers Squibb, Roche-Ventana, Boehringer-Ingelheim, Invitae (previously Archer Dx Inc - collaboration in minimal residual disease sequencing technologies) and Ono Pharmaceutical. He is an AstraZeneca Advisory Board member and Chief Investigator for the AZ MeRmaiD 1 and 2 clinical trials and is also Co-Chief Investigator of the NHS Galleri trial funded by GRAIL and a paid member of GRAIL's Scientific Advisory Board. He receives consultant fees from Achilles Therapeutics (also SAB member), Bicycle Therapeutics (also a SAB member), Genentech, Medicxi, Roche Innovation Centre – Shanghai, Metabomed (until July 2022), and the Sarah Canon Research Institute C.S. has received honoraria from Amgen, AstraZeneca, Pfizer, Novartis, GlaxoSmithKline, MSD, Bristol Myers Squibb, Illumina, and Roche-Ventana. C.S. had stock options in Apogen Biotechnologies and GRAIL until June 2021, and currently has stock options in Epic Bioscience, Bicycle Therapeutics, and has stock options and is co-founder of Achilles Therapeutics. C.S. holds patents relating to assay

technology to detect tumour recurrence (PCT/GB2017/053289); targeting neoantigens (PCT/EP2016/059401), identifying patent response to immune checkpoint blockade (PCT/EP2016/071471), determining HLA LOH (PCT/GB2018/052004), predicting survival rates of patients with cancer (PCT/GB2020/050221), identifying patients who respond to cancer treatment (PCT/GB2018/051912), US patent relating to detecting tumour mutations (PCT/US2017/28013), methods for lung cancer detection (US20190106751A1) and both a European and US patent related to identifying insertion/deletion mutation targets (PCT/GB2018/051892). J.D.B. has stock options in Tailor Bio and Inivata and is co-founder of Tailor Bio. J.D.B. has had consulting and advisory roles in AstraZeneca and Clovis Oncology and has received honoraria from GSK and Astrazeneca. J.D.B. holds patents relating to TAm-Seq v2 method for ctDNA estimation, enhanced detection of target DNA by fragment size analysis and methods for predicting treatment response in cancers. The remaining authors declare no competing interests.

## Additional information

**Filipe Correia Martins** [1,2,3,4,5,15] ✉, **Dominique-Laurent Couturier** [3,6,15], **Ines de Santiago** [3], **Carolin Margarethe Sauer** [3], **Maria Vias** [3], **Mihaela Angelova** [4], **Deborah Sanders** [3], **Anna Piskorz** [3], **James Hall** [3], **Karen Hosking** [7], **Anumithra Amirthanayagam** [5], **Sabina Cosulich** [8], **Larissa Carnevalli** [8], **Barry Davies** [8], **Thomas B. K. Watkins** [4], **Ionut G. Funingana** [3,9], **Helen Bolton** [5], **Krishnayan Haldar** [5], **John Latimer** [5], **Peter Baldwin** [5], **Robin Crawford** [5], **Matthew Eldridge** [3], **Bristi Basu** [7,9], **Mercedes Jimenez-Linan** [10], **Andrew W. Mcpherson** [11], **Nicholas McGranahan** [12], **Kevin Litchfield** [4], **Sohrab P. Shah** [11], **Iain McNeish** [13], **Carlos Caldas** [3,9], **Gerard Evan** [14], **Charles Swanton** [4,12] ✉ & **James D. Brenton** [3,9] ✉

[1]Department of Obstetrics and Gynaecology, University of Cambridge, Cambridge, UK. [2]Experimental Medicine Initiative, University of Cambridge, Cambridge, UK. [3]Cancer Research UK Cambridge Institute, University of Cambridge, Cambridge, UK. [4]Cancer Evolution and Genome Instability Laboratory, The Francis Crick Institute, London, UK. [5]Department of Gynaecological Oncology, Cambridge University Hospitals, Cambridge, UK. [6]Medical Research Council Biostatistics Unit, University of Cambridge, Cambridge, UK. [7]Cambridge University Hospitals, Cambridge, UK. [8]Early Oncology R&D, Astrazeneca, Cambridge, UK. [9]Department of Oncology, University of Cambridge, Cambridge, UK. [10]Department of Histopathology, Cambridge University Hospitals, Cambridge, UK. [11]Computational Oncology, Department of Epidemiology and Biostatistics, Memorial Sloan Kettering Cancer Centre, NYC, USA. [12]Cancer Research UK Lung Cancer Centre of Excellence, University College London Cancer Institute, London, UK. [13]Department of Surgery and Cancer, Imperial College of London, London, UK. [14]Department of Biochemistry, University of Cambridge, Cambridge, UK. [15]These authors contributed equally: Filipe Correia Martins, Dominique-Laurent Couturier. ✉e-mail: f.correiamartins@nhs.net; Charles.swanton@crick.ac.uk; james.brenton@cruk.cam.ac.uk

