## [Peer review file · Nature Communications]

REVIEWER COMMENTS

Reviewer #1 (Remarks to the Author):

My previous comments have been addressed

Reviewer #4 (Remarks to the Author):

I thank the authors for providing answers to some of my questions, but I am afraid that my overall impression of the manuscript has not substantially evolved after this revision. The changes to the manuscript are limited and largely appear to reflect lip service to various reviewer comments. My main concerns remain: the experimental design is not transparent; I could not understand the number of experiments and statistical tests that had been conducted to arrive at the (marginally) significant results presented in the manuscript. The authors justify which drug response associations to show based on prior literature, but it is unclear to me whether these were really the only drugs tested or whether there was a larger drug selection at some point and whether significant associations were chosen from that larger set? Were all 5 SCNAs of interest tested for association with every drug of interest and then only a subset presented? Overall, the manuscript remains a collection of various analyses that are only loosely collected logically and contain limited novelty.

As a side note, I find the authors' response to the question why cancer drivers should display a better correlation with copy number than other genes inscrutable. "We hypothesised that driver SCNAs would be selected owing to cellular survival advantage as opposed to passenger SCNA that would be randomly associated with gene expression changes." – SCNAs randomly associated with gene expression changes? What does this mean? It is well established that copy number alterations lead to predictable changes in chromosome-level gene expression, regardless of the identity of the genes on the chromosome, what does "randomly associated" mean? "Drivers are subject to selective advantages and therefore would be expected to have stronger correlations across cohorts" – Drivers are not subject to selective advantages, drivers *confer* selective advantage. However, this has nothing to do with the strength of correlation across any cohorts. Copy number events generate gene expression changes which subsequently may get selected for or not (depending on the competitive advantage they confer). The authors need to provide a much clearer (and theoretically backed) narrative on this issue in my opinion.

Regarding the response to reviewer 3, the authors have added effect sizes to the figure legends but the central concern that “the question whether SCNAs could be an important clinical biomarker has not been rigorously addressed” remains unaddressed.

Dear Dr. Gutierrez,

We were pleased that reviewer 1 was completely satisfied with our previous rebuttal. This document provides further responses to reviewer 4. The changes in the manuscript are highlighted in purple.

*I thank the authors for providing answers to some of my questions, but I am afraid that my overall impression of the manuscript has not substantially evolved after this revision. The changes to the manuscript are limited and largely appear to reflect lip service to various reviewer comments. **My main concerns remain: the experimental design is not transparent; I could not understand the number of experiments and statistical tests that had been conducted to arrive at the (marginally) significant results presented in the manuscript. The authors justify which drug response associations to show based on prior literature, but it is unclear to me whether these were really the only drugs tested or whether there was a larger drug selection at some point and whether significant associations were chosen from that larger set? Were all 5 SCNAs of interest tested for association with every drug of interest and then only a subset presented?***

Regarding our experimental design:

We thank the reviewer for pointing out that the description of the experimental design should be improved. Please note that one of the co-first authors of the MS (DLC) is a senior statistician.

We selected 12 drugs, four of which were representative of conventional chemotherapeutic agents and eight new targeted inhibitors. The 8 new compounds were selected based on their ability to impair DNA repair and the PI3K pathway, which are well known to be crucial for HGSOc survival. Despite only using the data related to a few of the 12 drugs (please see below), we still presented the whole dataset in the manuscript and we can confirm that these were the only drugs tested.

Before data analysis, based on the literature, we defined a list of 5 primary associations to be tested. These primary associations were based on gene-drug associations that had been well established in previous screens (e.g. high MYC – response to Paclitaxel; high KRAS – response to Doxorubicin) or that could be predicted based on the known functional impact of the specific SCNAs (high CCNE1 – resistance to pATM inhibitor; high MYC – response to mTOR inhibitor; high TERT - response to mTOR inhibitor). This resulted in the list of 5 *pre-specified* 'gene-drug' associations described in the table below:

	Gene	Drug	p-value 1	p-value 2
1	MYC	Paclitaxel	0.0491	0.0472
2	KRAS	Doxorubicin	0.1928	0.0410
3	MYC	AZD2014 (mTOR)	0.2593	0.0809
4	CCNE1	AZD0156 (ATM)	0.0438	0.1197
5	TERT	AZD2014 (mTOR)	0.1332	0.2228

1. p-value 1*: p-values of regression analyses considering gene expression as a continuous predictor,
2. p-value 2*: p-value of a Jonckheere's trend tests considering gene expression as an ordinal predictor.

The number of primary associations was determined based on a simulation study, performed prior to our experimental analysis that analysed the impacts of multiple testing and sampling error. This simulation considered 20 patients, 12 drugs and 5–10 copy number changes and showed that when

generating data under the null hypothesis, the absolute Pearson correlation estimates of all pairwise ‘drug and gene’ combinations would range on average between 0.0–0.6 (assuming independence between drug response and gene copy number). The probability of observing spurious relationships between gene copy number and drug response was hence too large when considering all pairwise combinations. *We therefore decided to only focus on promising associations based on the literature.* Annotations from our meetings and our R code show that we decided to select 5 (and no more than 10) signed associations prior to the analysis. Our assumptions were:

- **five**, a number small enough to strongly reduce the expected range of absolute correlation estimates under the null hypothesis and large enough to compare the distribution of association test p-values to the uniform via a Kolmogorov–Smirnov test (see below),
- **signed association**: the direction of the relationship was required to perform (more powerful) one-sided tests.

We apologise that this was not made clearer in the manuscript. There was no explanation on how the drugs were selected and no figure was available for the association between TERT copy number and AZD2014 inhibition. This has now been amended:

- **Section a/** of the Method Section is now dedicated to the experimental design of these analyses
- **Supplementary Figure F4F** now presents the association between AZD2014 drug response and TERT gene copy number (also shown below). The corresponding R code is available as supplemental material.

Figure 1 (Supplementary Figure S4F)

Regarding the level of significance of the 5 associations of interest:

Owing to our limited sample size, our p-values from individual gene-drug comparisons range from ‘borderline significance’ to non-significance, as the reviewer correctly points out. However, using a joint analysis, they show a strong support for the ability to predict drug response by means of gene copy number. Indeed, if there were no relationship between drug response and gene copy number, p-values would follow a uniform distribution as drug response is continuous (see also Murdoch et al 2008; ref 1). A Kolmogorov–Smirnov test could then be used to formally compare the empirical distribution function of our observed p-values with the uniform probability distribution.

Figure 2 (Supplementary Figure S6A)

The left plot of the figure 2 shows a QQ-plot comparing the observed ranked p-values available in the Table above (y-axis) to the corresponding uniform quantiles (x-axis) for two sets of analyses, respectively considering gene copy-number as a continuous and ordinal predictor of drug response (coloured lines). Kolmogorov–Smirnov test comparing the empirical distribution function of the p-values with the uniform probability distribution respectively led to p-values of 0.002 and 0.001 when considering copy number as a continuous and ordinal predictor.

The right plot shows the boxplot of p-values (on the log₁₀ scale) of the same Kolmogorov–Smirnov tests performed on 1000 *random* sets of 5 genes and considering gene copy-number as a continuous predictor. The pink horizontal line corresponds to log₁₀(0.002), the result observed when considering our pre-specified list of genes.

If there was no relationship between drug response and copy number, the p-values would be close to the unit line (dark blue) on the left plot, and the pink horizontal line would not correspond to an outlier on the right plot (only one random gene copy number - drug association over a 1000 had a smaller Kolmogorov-Smirnov p-value). We can note that this is far from being the case, thus showing a strong support for our hypothesis that copy number changes can predict drug response.

In this review, we also considered a larger list which includes all the associations discussed in the manuscript and that are either well recognised or supported by functional background knowledge (14/60 possible associations arising from 5 drugs × 12 copy number changes). Using the same statistical methodology, the same conclusion is reached with the larger list of associations. The left plot of the graph below shows that both ordinal and continuous analyses in observed p-values are significantly different from the Uniform distribution. Large p-values observed when considering copy number as an ordinal outcome correspond to cases in which the number of observations of some

levels is very small. The fact that 13 out of 14 p-values from the continuous are below 0.5 shows that the assumed direction of the relationship was correctly assessed for most hypotheses.

Figure 3 (Supplementary Figure S6B)

Figures 2 and 3 are now available in **Supplementary Figures 6 SA and SB** and the analyses related to the Kolmogorov-Smirnov tests and sensitivity analyses are described in **Section m/ of the Method Section**. The corresponding R code is available as supplemental material.

Overall, the manuscript remains a collection of various analyses that are only loosely collected logically and contain limited novelty.

Regarding the connection between the different analyses:

We politely disagree that our analyses are loosely connected and would lie to explain our reasoning. The figure below shows our main question, hypothesis and how our analyses are connected. It shows that we aimed to answer the question on how to predict drug response in HGSOC. We tested the hypothesis that allele copy number in driver genes associates with gene expression and that clonal/early driver alterations associate with drug response by defining relevant genomic biomarkers, by validating and using spheroids as a model to predict drug response and by using them to test associations between driver SCNAs and drug response. We further used genomic public datasets to validate an interesting association between MYC CN and mTOR inhibition and found co-occurrence of MYC SCNAs with SCNAs in PI3K genes.

Figure 4

Regarding the level of novelty of our study:

There is still considerable debate regarding the role of SCNAs in tumour evolution, and indeed whether selection is operating at all. There is very little in the literature regarding the clonality of SCNA events and their ability to confer therapeutic vulnerabilities on the emerging cancer clone. We believe that the following points are novel and worthy of a publication in *Nature Communications*:

- a: clonality analysis and definition of early/clonal HGSOc drivers
- b: preferential association between copy number and gene expression (possibly due to selection based on epigenetic/methylation features) for cancer drivers
- c: systematic association between copy number in clonal driver genes and response to conventional chemotherapeutic drugs and new targeted inhibitors
- d: validation of the association between MYC CN - paclitaxel response using clinical data in very well characterised cases.
- e: genomic validation of the association between MYC CN – response to mTOR inhibition across cohorts of HGSOc, triple negative breast cancer and squamous lung cancer.

As a side note, I find the authors' response to the question why cancer drivers should display a better correlation with copy number than other genes inscrutable. "We hypothesised that driver SCNAs would be selected owing to cellular survival advantage as opposed to passenger SCNA that would be randomly associated with gene expression changes." – **SCNAs randomly associated with gene expression changes? What does this mean? It is well established that copy number alterations lead to predictable changes in chromosome-level gene expression, regardless of the identity of the genes on the chromosome, what does "randomly associated" mean? "Drivers are subject to selective advantages and therefore would be expected to have stronger correlations across cohorts" – Drivers are not subject to selective advantages, drivers *confer* selective advantage.** However, this has nothing to do with the strength of correlation across any cohorts. Copy number events generate gene expression changes which subsequently may get selected for or not (depending on the competitive advantage they confer). The authors need to provide a much clearer (and theoretically backed) narrative on this issue in my opinion.

Many thanks for highlighting these points.

We apologise for the semantic errors in the rebuttal letter. By trying to provide more detail in the rebuttal, we ended up using expressions that were not as accurate as the ones used in the manuscript. It is widely assumed that SCNAs lead to predictable changes in the chromosome-level gene expression, regardless of the identity of the genes on the chromosome. However, exquisite selection effects can occur within amplicons. Recent evidence for this was provided by genomic characterization of the transcription factor *teashirt zinc finger homeobox 2 (TSHZ2)* gene which is specifically methylated within a frequently amplified locus in 20q13.2 in breast cancer [2]. These data show that gene function can determine gene expression (and selection against expression) in copy number aberrations. In this work, we hypothesised that:

1. the SCNA effect in gene expression is actually dependent of the identity of the genes;
2. SCNAs do not always result in the same gene expression changes due to selection based on epigenetic properties such as methylation.

For simplicity and to illustrate this concept, we can use an amplicon as an example. The oncogenes or putative drivers in an amplicon, will only confer a selective advantage if expressed, and therefore will tend to be hypomethylated and their expression tend to be highly correlated with the chromosomal copy number. In contrast, tumour suppressor genes in an amplicon will tend to be more hypermethylated and less expressed since if their increased copy number would translate in increased gene expression, this would confer a negative survival effect. Passenger genes that do not affect cell fitness or survival, would have a neutral in-between role – they can either be methylated or hypomethylated and with either strong or weak associations between copy number and gene expression without affecting how cells are selected. That’s what we meant about “randomly associated” but acknowledge that this was not clear in the rebuttal document; we have also edited the manuscript for further clarity on this point.

Regarding the response to reviewer 3, the authors have added effect sizes to the figure legends but the central concern that “the question whether SCNAs could be an important clinical biomarker has not been rigorously addressed” remains unaddressed.

We apologise for not addressing that point appropriately in the reviewer’s perspective. The aim of this work was not to validate individual clinical biomarkers. Instead we wanted to highlight that rigorous identification of early/clonal driver SCNAs and understanding how the gene expression changes in important driver genes impacted by the SCNA affect cell fitness, proliferation and survival, may provide important clues to understand drug response. Future work based on our work and awareness for the relevance of driver SCNAs (rather than focus on driver mutations) in chromosomally unstable tumours, may however lead into new clinical biomarkers of response to specific drugs. We have highlighted this both in the introduction and in the discussion to make it clear that the aim of this work was not to present clinically validated biomarkers of response.

REFERENCES

1. Murdoch, D, Tsai, Y, and Adcock, J (2008). P-Values are Random Variables. *The American Statistician*, 62, 242-245.
2. Uribe ML, Dahlhoff M, Batra RN, Nataraj NB, Haga Y, Drago-Garcia D, Marrocco I, Sekar A, Ghosh S, Vaknin I, Lebon S, Kramarski L, Tsutsumi Y, Choi I, Rueda OM, Caldas C, Yarden Y. TSHZ2 is an EGF-regulated tumor suppressor that binds to the cytokinesis regulator PRC1 and inhibits metastasis. *Sci Signal*. 2021;14(688).

REVIEWERS' COMMENTS

Reviewer #4 (Remarks to the Author):

I thank the authors for their replies. If the manuscript clearly states that the reported associations were pre-selected and not chosen post-hoc from a larger pool of analyses, a reader should be satisfied. The authors' argumentation regarding driver/passenger SCNAs and expression continues to evade me and appears speculative when it comes to the question of methylation, but I do not think that this point should hold back the paper.